# Iterative Trajectory Optimization for Physical-Layer Secure Buffer-Aided UAV Mobile Relaying

**DOI:** 10.3390/s19153442

**Published:** 2019-08-06

**Authors:** Lingfeng Shen, Ning Wang, Xiang Ji, Xiaomin Mu, Lin Cai

**Affiliations:** 1School of Information Engineering, Zhengzhou University, Zhengzhou 450001, China; 2Department of Electrical and Computer Engineering, University of Victoria, Victoria, BC V8W 2Y2, Canada

**Keywords:** buffer-aided relaying, physical-layer security, secrecy rate, trajectory optimization, UAV mobile relay

## Abstract

With the fast development of commercial unmanned aerial vehicle (UAV) technology, there are increasing research interests on UAV communications. In this work, the mobility and deployment flexibility of UAVs are exploited to form a buffer-aided relaying system assisting terrestrial communication that is blocked. Optimal UAV trajectory design of the UAV-enabled mobile relaying system with a randomly located eavesdropper is investigated from the physical-layer security perspective to improve the overall secrecy rate. Based on the mobility of the UAV relay, a wireless channel model that changes with the trajectory and is exploited for improved secrecy is established. The secrecy rate is maximized by optimizing the discretized trajectory anchor points based on the information causality and UAV mobility constraints. However, the problem is non-convex and therefore difficult to solve. To make the problem tractable, we alternatively optimize the increments of the trajectory anchor points iteratively in a two-dimensional space and decompose the problem into progressive convex approximate problems through the iterative procedure. Convergence of the proposed iterative trajectory optimization technique is proved analytically by the squeeze principle. Simulation results show that finding the optimal trajectory by iteratively updating the displacements is effective and fast converging. It is also shown by the simulation results that the distribution of the eavesdropper location influences the security performance of the system. Specifically, an eavesdropper further away from the destination is beneficial to the system’s overall secrecy rate. Furthermore, it is observed that eavesdropper being further away from the destination also results in shorter trajectories, which implies it being energy-efficient as well.

## 1. Introduction

Unmanned aerial vehicles (UAVs), also known as drones in many commercial applications, have witnessed a dramatic growth in the industry and market in the past few years. As the ecosystem is building up, there is an increasing research interest in UAV-related topics, in particular from the communication perspective. Due to their mobility and implementation flexibility, UAVs can be used as airborne mobile relays to assist terrestrial point-to-point communications where direct communication between the source and the destination is obstructed [1]. The Quality-of-Service (QoS) provisioning of the communication system can be significantly improved by jointly optimizing the UAV relay placement and the radio resource allocation [2,3]. UAV-based stations in the air may also be a viable means to solve the backhaul crunch that is critical to the deployment of dense small cell networks [4]. However, the use of UAVs as mobile relays also raises new problems and challenges to the communication system design. In particular, the characteristics of the air-to-ground wireless channels and the mobility of UAVs may bring in favorable conditions to potential eavesdropping and as such require new designs to better protect information security.

Because of the rapid development of computing power, the traditional cryptosystem based on computational security is facing continuing increasing challenges. Physical-layer (PHY) security has emerged as a promising supplement to the computational security because of its information-theoretic security nature [5,6]. In the context of wireless communications, PHY security technologies can achieve information-theoretic security by exploiting randomness in physical properties of wireless channels [7]. Reliable data transmissions with Shannon’s notion of *perfect secrecy* can be supported accordingly under realistic conditions over a wide range of wireless-channel models. In 1975, Wyner demonstrated the basic idea of PHY security with a noisy wire-tap channel model and illustrated that when the legitimate channel is more favorable than the eavesdropping channel, the communication between the source and the destination can achieve Shannon’s notion of perfect secrecy [8]. Since then, a number of wireless-channel models, e.g., broadcast channel, multiple access channel, relay channel, interference channel, have been studied from the PHY security perspective in noisy and interference communication environments. The PHY security of conventional cooperative relaying systems with fixed relay(s) has been extensively investigated in the literature [9,10,11,12]. The realizations of such schemes mainly rely on management and optimized allocation of the radio resources under static or quasi-static conditions, which cannot fully use the dynamic radio propagation environment from the spatial perspective to improve PHY security.

Recently, it has been demonstrated that UAV-assisted communications can adaptively change the UAV station’s position according to the dynamic radio propagation environment to better exploit the spatial degrees-of-freedom for performance improvement [13,14,15]. For instance, an altitude dependent model was proposed to conduct performance analysis for the power and sum-rate gains of UAV-based aerial base stations (ABSs) [13]. By adaptively changing the height of the ABS, optimization of the sum-rate or power can be achieved accordingly. Extending the above idea to the studies of PHY security related problems, UAV position can be exploited in the PHY security design to add an extra degree-of-freedom in the design variables such that improved security performance is expected. In [16], security challenges of UAV communications due to the dominant line-of-sight (LOS) transmission are identified, and possible solution approaches are envisioned from the PHY security perspective. In the case of air-to-ground communications, it is suggested that a well-designed UAV trajectory can be an effective means against terrestrial eavesdropping.

On the other hand, queue awareness and buffer-aided protocols have been shown, from the cross-layer design perspective, to also provide gains to the physical-layer performance of cooperative relaying communications [17,18,19,20]. UAV relays equipped with data storage can, therefore, benefit from both relay node mobility and buffer-aided relaying in a way that data packets can be stored and then transmitted at more favorable locations subject to certain QoS requirements. How such mechanism affects PHY security designs of UAV mobile relaying systems is an interesting problem that has yet been adequately studied.

Recent emerging research interests in UAV wireless communications have been mainly focused on resource allocation and trajectory optimization. A UAV-enabled data collection system for wireless sensor networks was considered in [21], where a shortest-tour trajectory design was proposed based on policy gradient reinforcement learning. Zeng et al. further considered joint source/relay transmit power allocation and mobile relay trajectory design in a throughput optimization problem for UAV-enabled mobile relaying systems, subject to practical mobility constraints of the UAV relay [22]. Similar works conducting joint UAV trajectory design and radio resource allocation have been reported for various system setups such as UAV-enabled wireless powered communication networks [23] and UAV-enabled amplify-and-forward relay networks [24]. In [25], the minimum average throughput of multiple users is maximized under delay considerations by jointly optimizing the UAV trajectory and OFDMA resource allocation. It can be observed that most of the existing works focus on the throughput performance. The PHY security aspect of the UAV mobile relaying has yet been adequately investigated. In [26], secrecy rate maximization was achieved by optimal power allocation at the source and the relay. Zhang et al. added the UAV trajectory to the design problem and studied maximization of the sum secrecy rate of the UAV by jointly designing the UAV trajectory and the transmit power control [27]. However, these works rely on a strong assumption of fixed and known eavesdropper location. More recently, multiple potential eavesdroppers with imperfect knowledge of the eavesdropper locations were considered in [28], where robust design of the UAV trajectory and the transmit power for PHY security optimization was investigated. Still, how UAV-enabled secure mobile relaying benefits from buffer-aided relaying is under-investigated.

In this work, PHY security of a buffer-aided UAV mobile relaying system is studied. Specifically, a four-node system model containing a source, a destination, a UAV mobile relay with finite data buffer, and a randomly located eavesdropper is considered. The sum secrecy rate of the system is maximized through UAV relay trajectory optimization. The main contributions of this work are summarized in the following.

Instead of making a strong assumption of known and static eavesdropper location/channel, in this work, a randomly located eavesdropper with only the statistical information of its location known to the legitimate system is considered in the secure trajectory design for buffer-aided UAV mobile relaying.By discretizing the total flight time into *N* equal quasi-static time slots and exploiting the buffer-aided relaying protocol, a sum secrecy rate maximization problem is formulated to find the optimal UAV relay trajectory anchor points that achieve the maximum sum secrecy rate.The lower bounds of the maximal achievable rates are derived through Taylor’s expansion. The accuracy of the lower bounding technique is guaranteed by extra upper bounding the rates in the constraints of the optimization problem.To make the original non-convex problem tractable, an iterative trajectory optimization scheme is proposed. Specifically, instead of optimizing the trajectory anchor points of the UAV directly, the increments from the previous iteration for each anchor point are iteratively optimized. The problem is then decomposed into successive convex approximation subproblems by invoking the rate bounds in an iterative procedure. The convergence of this trajectory iteration method is proved analytically by the squeeze principle.

Simulation results illustrate that the method of finding the optimal trajectory by iterative incrementing of the anchor points is effective and fast converging. The simulation results show that the trajectory of the UAV converges in around 10 iterations, and the performance of the system’s sum secrecy rate is significantly improved. The location of the eavesdropper affects the security performance of the system. Specifically, the eavesdropper further away from the destination is more favorable to the system’s secrecy capacity. Furthermore, it was observed that having higher maximum UAV speed is also beneficial to the improvement of the secrecy rate performance.

The remainder of this paper is organized as follows. In Section 2, we present the buffer-aided UAV relaying system model and give an initial description of the trajectory optimization problem. In Section 3, the solution approach based on decomposition and progressive convex approximation of the original non-convex problem is proposed. Three propositions are presented, and we prove analytically the trajectory iteration method converges. Simulation results are presented in Section 4, and concluding remarks are made in Section 5.

## 2. System Model and Problem Description

A UAV mobile relaying wireless communication system model as shown in Figure 1 is considered in this work. There are four single-antenna nodes in the model: a single source (S), a single destination (D), a UAV mobile relay (R), and an eavesdropper (E). Suppose the source and the destination are fixed in a straight line on the ground, which is designated as the $d_{x}$ axis in the model. The positions of the source and the destination in the two-dimensional (2D) space are denoted by $\left(L_{s},0\right)$ and $\left(L_{d},0\right)$, respectively. The ground-based eavesdropper is located at $\left(L_{e},0\right)$. In this work, it is assumed that $L_{e}$ is a random variable, and uniformly distributed $L_{e}$ is considered in the subsequent analysis to demonstrate the proposed solution approach. Specifically, $L_{e}$ is uniformly distributed in an interval [a,b], where *a* and *b* are two real valued constants with a≤b. This work and the proposed solution technique can be extended to scenarios with more complex geometries of the node locations. Direct communication between the source and the destination is assumed to be blocked. In addition, it is assumed that the eavesdropper cannot receive direct transmissions from the source, either. The UAV moves in the 2D geographical area at a fixed height *h* above the terrestrial communication system to assist communications between the source and the destination. It also raises information security issues because the ground-based eavesdropper can now receive the forwarded signals from the UAV relay.

Ignoring the taking off and landing processes, the UAV serves as a mobile relay for a finite time horizon *T*, and its starting and ending points are denoted as **SP** and **EP**, respectively, as shown in Figure 1. For convenience, we designate the location of **SP** as the origin, and the location of **EP** is denoted as (L,0). As the UAV moves, the distance between the UAV and each terminal is constantly changing, and the channel gains of the corresponding communication links change accordingly. A dynamic channel model is established to reflect these changes with the UAV location. The UAV relay’s service time interval *T* is divided into *N* equally spaced time slots. Each time slot is sufficiently short to guarantee the quasi-static assumption, i.e., the wireless channels are almost constant within one time slot. The *N* time slots then correspond to *N* decision instants for the trajectory, and the UAV position $\left(d_{x}\left[n\right],d_{y}\left[n\right]\right)$ at the beginning of the *n*th time slot is used to characterize the wireless channels of the corresponding decision instant. Based on the above assumptions on the starting and ending points, there are $(d_{x}\left[1\right],d_{y}\left[1\right])=\left(0,0\right)$ and $(d_{x}\left[N+1\right],d_{y}\left[N+1\right])=\left(L,0\right)$. The UAV relay R operates in a time-division duplex (TDD) mode, with equal time allocation for the S-R transmission and the R-D transmission. A finite data buffer of size *B* is equipped by the UAV relay to enable buffer-aided relaying. The channel coefficients of the S-R, R-D, and R-E channels in time slot *n* are denoted by $h_{sr}\left[n\right]$, $h_{rd}\left[n\right]$, and $h_{re}\left[n\right]$, respectively. It is assumed that the UAV relay is flying at a height where the path is clear of obstacles to allow total freedom in the trajectory design. This requires the UAV to fly at a relatively high altitude to be well above all the buildings. Consequently, as discussed in [13], the LOS path dominates the air-to-ground channel. The large-scale free-space path-loss is the dominating factor in $h_{sr}\left[n\right]$, $h_{rd}\left[n\right]$, and $h_{re}\left[n\right]$. The S-R channel path-loss is given as [29]
(1)$$\begin{array}{cc} PL_{sr}\left[n\right] & =PL_{sr}\left(d_{0}\right)+10\bar{n}\log\left(d_{sr}\left[n\right]/d_{0}\right),\;\;n=1,\ldots ,N, \end{array}$$
where $d_{0}$ is the free-space reference distance, and *d* is the distance between the transmitter and the receiver. A path-loss exponent $\bar{n}=2$ is used due to the large elevation angle of the air-to-ground communication system model under consideration [13]. As a result, (Equation 1) is written as
(2)$$\begin{array}{cc} PL_{sr}\left[n\right] & =C_{0}+20\log\left(d_{sr}\left[n\right]\right),\;\;n=1,\ldots ,N, \end{array}$$
where $C_{0}=PL_{sr}\left(d_{0}\right)-20\log\left(d_{0}\right)$. Let $C=10^{C_{0}/10}$, the large-scale S-R channel coefficient in time slot *n* is approximately given as
(3)$$h_{sr}\left[n\right]=\frac{1}{C\left(h^{2}+{\left(d_{x}\left[n\right]-L_{s}\right)}^{2}+d_{y}^{2}\left[n\right]\right)},\;\;n=1,\ldots ,N.$$

The approximate channel coefficients of the S-R and R-D channels can be obtained similarly as
(4)$$h_{rd}\left[n\right]=\frac{1}{C\left(h^{2}+{\left(d_{x}\left[n\right]-L_{d}\right)}^{2}+d_{y}^{2}\left[n\right]\right)},\;\;n=1,\ldots ,N,$$
and
(5)$$h_{re}\left[n\right]=\frac{1}{C\left(h^{2}+{\left(d_{x}\left[n\right]-L_{e}\right)}^{2}+d_{y}^{2}\left[n\right]\right)},\;\;n=1,\ldots ,N.$$

As the UAV relay moves in the 2D geographic area in the air, the wireless-channel states constantly change, resulting in different $h_{sr}\left[n\right]$, $h_{rd}\left[n\right]$, and $h_{re}\left[n\right]$ values in different time slots. The corresponding achievable rate and secrecy rate also change accordingly. In contrast to conventional wireless communication systems where the channel coefficients’ changes with time are mainly due to fading that has a random nature, in the UAV mobile relaying system studied in this work, based on the above assumptions of the air-to-ground channels, these changes are primarily determined by the UAV trajectory and therefore can be planned ahead, in an off-line manner. It is then possible to improve the sum achievable secrecy rate of the UAV mobile relaying system by designing a favorable UAV trajectory. The computation task of finding the optimal trajectory, as a result, can be offloaded to a ground-based computing facility with controllable communication overhead considering the limited computing power and battery lifetime of the UAV relay. This is important to the practical implementation of the proposed design technique.

Denote by $R_{s}\left[n\right]$ and $R_{d}\left[n\right]$ the maximum achievable rates of the S-R and R-D channels in the *n*th time slot. It is straightforward to show that
(6)$$\begin{array}{cc} R_{s}\left[n\right] & =\log_{2}\left(1+\frac{p_{s}h_{sr}\left[n\right]}{WN_{0}}\right)\\ & =\log_{2}\left(1+\frac{p_{s}}{CWN_{0}\left(h^{2}+{\left(d_{x}\left[n\right]-L_{s}\right)}^{2}+d_{y}^{2}\left[n\right]\right)}\right), \end{array}$$
and
(7)$$\begin{array}{cc} R_{d}\left[n\right] & =\log_{2}\left(1+\frac{p_{r}h_{rd}\left[n\right]}{WN_{0}}\right)\\ & =\log_{2}\left(1+\frac{p_{r}}{CWN_{0}\left[h^{2}+{\left(d_{x}\left[n\right]-L_{d}\right)}^{2}+d_{y}^{2}\left[n\right]\right]}\right), \end{array}$$
where $p_{s}$ and $p_{r}$ represent the transmit power of the source and the UAV relay, respectively, *W* is the communication bandwidth, and $N_{0}$ is the power spectral density of the additive white Gaussian noise (AWGN). Because only statistical information about the eavesdropper location is known to the legitimate communication system, and the UAV position keeps changing along time, the eavesdropper’s ergodic achievable rate in the *n*th time slot, denote by $R_{e}\left[n\right]$, is a reasonable measure of the eavesdropper capability. By definition of ergodic rate, $R_{e}\left[n\right]$ is the expected value of the R-E rate over the distribution of the eavesdropper location.
(8)$$\begin{array}{cc} R_{e}\left[n\right] & =E\left[\log_{2}\left(1+\frac{p_{r}h_{re}\left[n\right]}{WN_{0}}\right)\right]\\ & =E\left[\log_{2}\left(1+\frac{p_{r}}{CWN_{0}\left[h^{2}+{\left(d_{x}\left[n\right]-L_{e}\right)}^{2}+d_{y}^{2}\left[n\right]\right]}\right)\right]. \end{array}$$

The idea of PHY security is based on the notion of perfect secrecy, which requires the information leaked about the transmitted message to the eavesdropper is asymptotically zero. Maximal achievable secrecy rate, or secrecy capacity, characterizes the maximal rate at which the legitimate receiver can reliably recover the message, while the eavesdropper obtains no information about the message. The underlying idea is that the existence of the eavesdropper undermines the reliable transmission between the legitimate parties from information security perspective. The mutual information between the legitimate parties is penalized by the amount of the mutual information of the transmitter-eavesdropper link. Conditioned on the quasi-static fading in one time slot, the second-hop (R-D/R-E) channel can be modeled as a discrete memoryless AWGN wire-tap channel. The corresponding ergodic secrecy rate in the *n*th time slot is then given as $R^{*}\left[n\right]={\left[R_{d}\left[n\right]-R_{e}\left[n\right]\right]}^{+}$, where ${\left[x\right]}^{+}=\max\left\{x,0\right\}$. To improve PHY security in the trajectory design, the following optimization problem P1 is formulated that maximizes the sum ergodic secrecy rate by finding the optimal UAV trajectory points $\left(d_{x}\left[n\right],d_{y}\left[n\right]\right)$ for all n=2,…,N.
(9)$$\;P1:\;\;\underset{{\left\{d_{x}\left[n\right],d_{y}\left[n\right]\right\}}_{n=2}^{N}}{\mathrm{maximize}}\;\sum_{n=1}^{N}R^{*}\left[n\right]$$
(9a)$$\begin{array}{cc} \;s.t.\; & \sum_{i=1}^{n}R^{*}\left[n\right]\leq \sum_{i=1}^{n}R_{s}\left[n\right]+B,\;\;\;n=1,\ldots ,N; \end{array}$$
(9b)$$\begin{array}{cc} \; & {\left(d_{x}\left[n+1\right]-d_{x}\left[n\right]\right)}^{2}+{\left(d_{y}\left[n+1\right]-d_{y}\left[n\right]\right)}^{2}\leq v^{2},\;n=1,\ldots ,N; \end{array}$$
where *v* and *B* represent the UAV’s maximum speed and buffer size, respectively. Equation ([Disp-formula FD9a-sensors-19-03442]) is the information causality and buffer size constraint for buffer-aided relaying, which implies that the forwarded secrecy packets must be cached in a buffer of size no larger than *B*. And ([Disp-formula FD9b-sensors-19-03442]) sets constraints on the UAV’s mobility, taking into consideration both the UAV’s starting and ending locations as well as the maximum UAV speed. Owing to the form of the objective function and the information causality constraint ([Disp-formula FD9a-sensors-19-03442]), it can be shown that the original problem P1 is non-convex. In the following section, we reformulate P1 by change of variables and successive convex approximation to make the problem mathematically tractable.

## 3. The Progressive Convex Approximation Method for the Non-Convex Problem

In this section, firstly the design variables are changed to transform the original problem P1 into a more friendly form. An iterative updating procedure of the trajectory anchor points based on optimization of the increments of each anchor point in each iteration is proposed. Lower bounding the rate expressions in each algorithm iteration by Taylor’s expansion results in convex subproblems which can be readily solved by standard techniques for convex optimization. This successive convex approximation procedure is shown to approach the optimal trajectory progressively with good convergence properties. The optimality gap of the proposed iterative optimization technique is shown to be very small with only a few algorithm iterations.

### 3.1. Change of Variables and Lower Bounding the Achievable Rates

It can be observed from Problem P1 that optimizing the trajectory anchor points $\left(d_{x}\left[n\right],d_{y}\left[n\right]\right)$ directly is cumbersome due to the analytic forms of the objective function and the constraints. Alternatively, because of the assumption of linear motion between decision (anchor) points, we propose to optimize the trajectory increments for each anchor point, denoted (η[n]≥0,ξ[n]≥0), in an iterative procedure. The results have shown that finding the optimal trajectory through optimizing the increments is effective and fast converging.

Assume the trajectory increment on the *n*th trajectory anchor point obtained in the *l*th algorithm iteration is $\left\{\eta^{\left(l\right)}\left[n\right],\xi^{\left(l\right)}\left[n\right]\right\}$, n=0,1,…,N. By setting an initial trajectory, e.g., the straight line segment from the source to the destination, it is straightforward to obtain the corresponding initial values of the anchor points, i.e., $\left\{\left(d_{x}^{\left(0\right)}\left[n\right],d_{y}^{\left(0\right)}\left[n\right]\right)\right\}$. The trajectory anchor points for the *l*th algorithm iteration is updated after the (l−1)th algorithm iteration as
(10a)$$\begin{array}{cc} d_{x}^{\left(l\right)}\left[n\right] & =d_{x}^{\left(l-1\right)}\left[n\right]+\eta^{\left(l-1\right)}\left[n\right], \end{array}$$
(10b)$$\begin{array}{cc} d_{y}^{\left(l\right)}\left[n\right] & =d_{y}^{\left(l-1\right)}\left[n\right]+\xi^{\left(l-1\right)}\left[n\right]. \end{array}$$

The achievable rate of the S-R channel in the *l*th algorithm iteration is calculated as
(11)$$R_{s}^{\left(l\right)}\left[n\right]\overset{}{≜}\log_{2}\left(1+\frac{p_{s}h_{sr}^{\left(l\right)}\left[n\right]}{WN_{0}}\right),$$
where the channel coefficient $h_{sr}^{\left(l\right)}\left[n\right]$ is calculated based on $\left(d_{x}^{\left(l\right)}\left[n\right],d_{y}^{\left(l\right)}\left[n\right]\right)$. Similarly, the achievable rates of the R-D and R-E channels for the current iteration, denoted $R_{d}^{\left(l\right)}\left[n\right]$ and $R_{e}^{\left(l\right)}\left[n\right]$, can also be obtained. The *l*th iteration is then an optimization problem about the trajectory point increments $\left\{\left(\eta^{\left(l\right)}\left[n\right],\xi^{\left(l\right)}\left[n\right]\right)\right\}$.
(12)$$\;P1^{\left(l\right)}:\;\;\underset{{\left\{\left(\eta^{\left(l\right)}\left[n\right],\xi^{\left(l\right)}\left[n\right]\right)\right\}}_{n=1}^{N}}{\mathrm{maximize}}\;\sum_{n=1}^{N}R^{*(l+1)}\left[n\right]$$
(12a)$$\begin{array}{cc} \;s.t.\; & \sum_{i=1}^{n}R^{*(l+1)}\left[n\right]\leq \sum_{i=1}^{n}R_{s}^{\left(l+1\right)}\left[n\right]+B,\;\;\;n=1,\ldots ,N; \end{array}$$
(12b)$$\begin{array}{cc} \; & {\left(d_{x}^{\left(l\right)}\left[1\right]+\eta^{\left(l\right)}\left[1\right]\right)}^{2}+{\left(d_{y}^{\left(l\right)}\left[1\right]+\xi^{\left(l\right)}\left[1\right]\right)}^{2}\leq v^{2}; \end{array}$$
$$\begin{array}{cc} \; & {\left(d_{x}^{\left(l\right)}\left[n+1\right]+\eta^{\left(l\right)}\left[n+1\right]-d_{x}^{\left(l\right)}\left[n\right]-\eta^{\left(l\right)}\left[n\right]\right)}^{2} \end{array}$$
(12c)$$\begin{array}{cc} \; & +{\left(d_{y}^{\left(l\right)}\left[n+1\right]+\xi^{\left(l\right)}\left[n+1\right]-d_{y}^{\left(l\right)}\left[n\right]-\xi^{\left(l\right)}\left[n\right]\right)}^{2}\leq v^{2},\;n=1,\ldots ,N-1; \end{array}$$
(12d)$$\begin{array}{cc} \; & {\left(d_{x}^{\left(l\right)}\left[N\right]+\eta^{\left(l\right)}\left[N\right]-L\right)}^{2}+{\left(d_{y}^{\left(l\right)}\left[N\right]+\xi^{\left(l\right)}\left[N\right]\right)}^{2}\leq v^{2}. \end{array}$$

The iterative procedure that updates $\left\{\left(d_{x}^{\left(l\right)}\left[n\right],d_{y}^{\left(l\right)}\left[n\right]\right)\right\}$ and $\left\{\left(\eta^{\left(l\right)}\left[n\right],\xi^{\left(l\right)}\left[n\right]\right)\right\}$ alternatively is conducted until some convergence criteria are met.

The subproblem $P1^{\left(l\right)}$ of the *l*th iteration obtained after the conversion is still non-convex. In order to deal with the non-convexity in the rate expressions in $P1^{\left(l\right)}$, a lower bounding technique based on Taylor’s expansion is proposed. The idea of the proposed technique is illustrated through three propositions in the following.

**Proposition** **1.**
*For any trajectory increment $\left(\eta^{\left(l\right)}\left[n\right],\xi^{\left(l\right)}\left[n\right]\right)$, the inequalities below must hold for all algorithm iterations.*
(13)$$\begin{array}{cc} R_{s}^{\left(l+1\right)}\left[n\right]\geq R_{s}^{(l+1)lb}\left[n\right]\overset{}{≜}\; & R_{s}^{\left(l\right)}\left[n\right]-a_{s}^{\left(l\right)}\left[n\right]\left[{\left(\eta^{\left(l\right)}\left[n\right]\right)}^{2}+{\left(\xi^{\left(l\right)}\left[n\right]\right)}^{2}\right]\\ & -b_{s}^{\left(l\right)}\left[n\right]\eta^{\left(l\right)}\left[n\right]-c_{s}^{\left(l\right)}\left[n\right]\xi^{\left(l\right)}\left[n\right], \end{array}$$
(14)$$\begin{array}{cc} R_{d}^{\left(l+1\right)}\left[n\right]\geq R_{d}^{(l+1)lb}\left[n\right]\overset{}{≜} & R_{d}^{\left(l\right)}\left[n\right]-a_{d}^{\left(l\right)}\left[n\right]\left[{\left(\eta^{\left(l\right)}\left[n\right]\right)}^{2}+{\left(\xi^{\left(l\right)}\left[n\right]\right)}^{2}\right]\\ & -b_{d}^{\left(l\right)}\left[n\right]\eta^{\left(l\right)}\left[n\right]-c_{d}^{\left(l\right)}\left[n\right]\xi^{\left(l\right)}\left[n\right], \end{array}$$
*where $a_{s}^{\left(l\right)}\left[n\right]$, $a_{d}^{\left(l\right)}\left[n\right]$, $b_{s}^{\left(l\right)}\left[n\right]$, $b_{d}^{\left(l\right)}\left[n\right]$, $c_{s}^{\left(l\right)}\left[n\right]$ and $c_{d}^{\left(l\right)}\left[n\right]$ are coefficients given in the following proof.*


**Proof.** Firstly, we define the function form
(15)$$f\left(Z\right)\overset{}{≜}\log_{2}\left(1+\frac{\lambda }{A+Z}\right),$$
with constants λ>0 and *A*. When Z>−A, f(Z) is convex, i.e., $f\left(Z\right)\geq f\left(Z_{0}\right)+f^{{}^{\prime }}\left(Z_{0}\right)\left(Z-Z_{0}\right)$ for any feasible $Z_{0}$. The achievable rate of the S-R channel in the (l+1)th algorithm iteration is given as
(16)$$\begin{array}{cc} R_{s}^{\left(l+1\right)}\left[n\right] & =\log_{2}\left(1+\frac{p_{s}h_{sr}^{\left(l+1\right)}\left[n\right]}{WN_{0}}\right)\\ & =\log_{2}\left(1+\frac{p_{s}}{CWN_{0}\left[h^{2}+{\left(d_{x}^{\left(l+1\right)}\left[n\right]-L_{s}\right)}^{2}+{\left(d_{y}^{\left(l+1\right)}\left[n\right]\right)}^{2}\right]}\right), \end{array}$$
where $d_{x}^{\left(l+1\right)}\left[n\right]=d_{x}^{\left(l\right)}\left[n\right]+\eta^{\left(l\right)}\left[n\right]$ and $d_{y}^{\left(l+1\right)}\left[n\right]=d_{y}^{\left(l\right)}\left[n\right]+\xi^{\left(l\right)}\left[n\right]$. Equation (Equation 16) can be fitted into the form of (Equation 15) with the coefficients given by
(17a)$$\begin{array}{cc} \;\lambda_{s} & =\frac{p_{s}}{CWN_{0}}, \end{array}$$
(17b)$$\begin{array}{cc} \;A_{s} & =h^{2}+{\left(d_{x}^{\left(l\right)}\left[n\right]-L_{s}\right)}^{2}+{\left(d_{y}^{\left(l\right)}\left[n\right]\right)}^{2}, \end{array}$$
(17c)$$\begin{array}{cc} Z_{s} & ={\left(\eta^{\left(l\right)}\left[n\right]\right)}^{2}+{\left(\xi^{\left(l\right)}\left[n\right]\right)}^{2}+2\left(d_{x}^{\left(l\right)}\left[n\right]-L_{s}\right)\eta^{\left(l\right)}\left[n\right]+2d_{y}^{\left(l\right)}\left[n\right]\xi^{\left(l\right)}\left[n\right]. \end{array}$$
It is straightforward to show that $Z_{s}>-A_{s}$ for $R_{s}^{\left(l+1\right)}\left[n\right]$. Therefore, by convexity $f\left(Z\right)\geq f\left(Z_{0}\right)+f^{{}^{\prime }}\left(Z_{0}\right)\left(Z-Z_{0}\right)$. Let $Z_{0}=0$, it can be shown that
$$f\left(Z_{0}\right)=\log_{2}\left(1+\frac{\lambda_{s}}{A_{s}}\right),$$
and
$$f^{{}^{\prime }}\left(Z_{0}\right)=-\frac{\lambda_{s}}{\ln2\left(A_{s}+\lambda_{s}\right)A_{s}}.$$
As a result,
$$\begin{array}{cc} R_{s}^{\left(l+1\right)}\left[n\right]\geq & R_{s}^{\left(l\right)}\left[n\right]-a_{s}^{\left(l\right)}\left[n\right]\left[{\left(\eta^{\left(l\right)}\left[n\right]\right)}^{2}+{\left(\xi^{\left(l\right)}\left[n\right]\right)}^{2}\right]\\ & -b_{s}^{\left(l\right)}\left[n\right]\eta^{\left(l\right)}\left[n\right]-c_{s}^{\left(l\right)}\left[n\right]\xi^{\left(l\right)}\left[n\right], \end{array}$$
where
(18a)$$\begin{array}{cc} \;a_{s}^{\left(l\right)}\left[n\right] & =\frac{\lambda_{s}}{\ln2\left(A_{s}+\lambda_{s}\right)A_{s}}, \end{array}$$
(18b)$$\begin{array}{cc} \;b_{s}^{\left(l\right)}\left[n\right] & =2\left(d_{x}^{\left(l\right)}\left[n\right]-L_{s}\right)a_{s}^{\left(l\right)}\left[n\right], \end{array}$$
(18c)$$\begin{array}{cc} c_{s}^{\left(l\right)}\left[n\right] & =2d_{y}^{\left(l\right)}\left[n\right]a_{s}^{\left(l\right)}\left[n\right]. \end{array}$$
Lower bound of the R-D channel rate $R_{d}^{\left(l+1\right)}\left[n\right]$ in (Equation 14) can be obtained in the same way, with
(19a)$$\begin{array}{cc} \;a_{d}^{\left(l\right)}\left[n\right] & =\frac{\lambda_{d}}{\ln2\left(A_{d}+\lambda_{d}\right)A_{d}}, \end{array}$$
(19b)$$\begin{array}{cc} \;b_{d}^{\left(l\right)}\left[n\right] & =2\left(d_{x}^{\left(l\right)}\left[n\right]-L_{d}\right)a_{d}^{\left(l\right)}\left[n\right], \end{array}$$
(19c)$$\begin{array}{cc} c_{d}^{\left(l\right)}\left[n\right] & =2d_{y}^{\left(l\right)}\left[n\right]a_{d}^{\left(l\right)}\left[n\right]. \end{array}$$
The coefficients $\lambda_{d}$ and $A_{d}$ in ([Disp-formula FD19a-sensors-19-03442]) are given by
(20a)$$\begin{array}{cc} \;\lambda_{d} & =\frac{p_{r}}{CWN_{0}}, \end{array}$$
(20b)$$\begin{array}{cc} A_{d} & =h^{2}+{\left(d_{x}^{\left(l\right)}\left[n\right]-L_{d}\right)}^{2}+{\left(d_{y}^{\left(l\right)}\left[n\right]\right)}^{2}. \end{array}$$
This completes the proof. □

Unlike the source and the destination, the location of the eavesdropper is assumed to be random and follows a uniform distribution. Lower bounding the eavesdropper rate therefore needs to be done in a slightly different way compared with $R_{s}$ and $R_{d}$ in Proposition 1. Instead, we give the following Proposition 2 about the ergodic eavesdropper rate that takes into consideration the distribution of the eavesdropper location.

**Proposition** **2.**
*The following inequality must hold for any trajectory increment $\left(\eta^{\left(l\right)}\left[n\right],\xi^{\left(l\right)}\left[n\right]\right)$*
(21)$$\begin{array}{cc} R_{e}^{\left(l+1\right)}\left[n\right]\geq R_{e}^{(l+1)lb}\left[n\right]\overset{}{≜} & R_{s}^{\left(l\right)}\left[n\right]-E\left[a_{s}^{\left(l\right)}\left[n\right]\right]\left({\left(\eta^{\left(l\right)}\left[n\right]\right)}^{2}+{\left(\xi^{\left(l\right)}\left[n\right]\right)}^{2}\right)\\ & -E\left[b_{s}^{\left(l\right)}\left[n\right]\right]\eta^{\left(l\right)}\left[n\right]-E\left[c_{s}^{\left(l\right)}\left[n\right]\right]\xi^{\left(l\right)}\left[n\right], \end{array}$$
*where $E\left[a_{s}^{\left(l\right)}\left[n\right]\right]$, $E\left[b_{s}^{\left(l\right)}\left[n\right]\right]$, and $E\left[c_{s}^{\left(l\right)}\left[n\right]\right]$ are coefficients given in the proof detailed in Appendix A.*


**Proof.** Please see Appendix A. □

### 3.2. Convergence of the Iterative Trajectory Optimization Technique

In this subsection, we first show the accuracy of the lower bounds on the rate expressions obtained in Section 3.1, which is important to the validity of the proposed iterative optimization algorithm. To guarantee validity and accuracy of the lower bounding technique proposed in Section 3.1, the following two additional inequality constraints on the lower bounds are introduced to the optimization problem.
(22a)$$\begin{array}{cc} R_{d}^{(l+1)lb}\left[n\right] & \geq R_{d}^{\left(l+1\right)}\left[n\right], \end{array}$$
(22b)$$\begin{array}{cc} R_{e}^{(l+1)lb}\left[n\right] & \geq R_{e}^{\left(l+1\right)}\left[n\right]. \end{array}$$

Combining the above inequalities with Propositions 1 and 2, we have
(23a)$$\begin{array}{cc} R_{d}^{(l+1)lb}\left[n\right] & \geq R_{d}^{\left(l+1\right)}\left[n\right]\geq R_{d}^{(l+1)lb}\left[n\right], \end{array}$$
(23b)$$\begin{array}{cc} R_{e}^{(l+1)lb}\left[n\right] & \geq R_{e}^{\left(l+1\right)}\left[n\right]\geq R_{e}^{(l+1)lb}\left[n\right]. \end{array}$$

By the squeeze principle, it is straightforward that equalities must hold for any feasible solution to the optimization problem adopting the additional constraints. As a result,
$$\begin{array}{cc} R_{d}^{(l+1)lb}\left[n\right] & =R_{d}^{\left(l+1\right)}\left[n\right],\\ R_{e}^{(l+1)lb}\left[n\right] & =R_{e}^{\left(l+1\right)}\left[n\right]. \end{array}$$

In the meantime, adding constraints ([Disp-formula FD22a-sensors-19-03442]) and ([Disp-formula FD22b-sensors-19-03442]) to the optimization problem with the above lower bounding technique can also guarantee convergence of the trajectory iteration. Next, we show through the following Proposition 3 convergence of the proposed iterative trajectory optimization technique.

**Proposition** **3.**
*The sum secrecy rate of the UAV relay system converges if the following inequalities must hold.*
$$\begin{array}{cc} R_{d}\left[n\right] & \leq R_{d}^{(l+1)lb}\left[n\right],\\ R_{e}\left[n\right] & \leq R_{e}^{(l+1)lb}\left[n\right]. \end{array}$$


**Proof.** For convenience, in the following proof the iteration index *l* and the time index *n* are omitted because the general results apply to all the trajectory points.If the inequalities ([Disp-formula FD22a-sensors-19-03442]) and ([Disp-formula FD22b-sensors-19-03442]) must hold, then as in (23) there have $R_{d}\geq R_{d}^{lb}\geq R_{d}$ and $R_{e}\geq R_{e}^{lb}\geq R_{e}$. By definition, the secrecy rate to be maximized is $R^{*}={\left[R_{d}-R_{e}\right]}^{+}$. Consequently, there must be $R_{d}^{lb}-R_{e}^{lb}\geq R^{*}\geq R_{d}^{lb}-R_{e}^{lb}$.In Section 3.1, it is assumed that the optimization variables are nonnegative, i.e., η≥0, ξ≥0. A positive pair (η,ξ) should always be found in an algorithm iteration, which leads to an improved secrecy rate until both η and ξ are zero. The secrecy rate is thus non-decreasing over the iterations.Hence, $R^{*}$ is monotonically increasing and bounded with respect to the optimization variables η and ξ. The convergence of the proposed iterative optimization method is thus proved. □

Based on the above discussions, the original Problem P1 can be accurately solved through an iterative procedure as described in Section 3.1 by solving the following constrained Problem $P2^{\left(l\right)}$ in each algorithm iteration until convergence.
(24)$$\;P2^{\left(l\right)}:\;\underset{\begin{array}{c} \begin{array}{c} {\left\{\left(\xi \left[n\right],\eta \left[n\right]\right)\right\}}_{n=1}^{N},\\ {\left\{R_{d}^{\left(l+1\right)}\left[n\right],R_{e}^{\left(l+1\right)}\left[n\right]\right\}}_{n=1}^{N} \end{array} \end{array}}{\mathrm{maximize}}\sum_{n=1}^{N}R^{*(l+1)}\left[n\right]$$
(24a)$$\begin{array}{cc} \;s.t.\; & \sum_{i=1}^{n}R^{*(l+1)}\left[i\right]\leq \sum_{i=1}^{n}R_{s}^{(l+1)lb}\left[i\right]+B,\;n=1,\ldots ,N; \end{array}$$
(24b)$$\begin{array}{cc} \; & R_{d}^{\left(l+1\right)}\left[n\right]\leq R_{d}^{(l+1)lb}\left[n\right],\;n=1,\ldots ,N; \end{array}$$
(24c)$$\begin{array}{cc} \; & R_{e}^{\left(l+1\right)}\left[n\right]\leq R_{e}^{(l+1)lb}\left[n\right],\;n=1,\ldots ,N; \end{array}$$
(24d)$$\begin{array}{cc} \; & {\left(d_{x}^{\left(l\right)}\left[1\right]+\eta^{\left(l\right)}\left[1\right]\right)}^{2}+{\left(d_{y}^{\left(l\right)}\left[1\right]+\xi^{\left(l\right)}\left[1\right]\right)}^{2}\leq v^{2}; \end{array}$$
$$\begin{array}{cc} \; & {\left(d_{x}^{\left(l\right)}\left[n+1\right]+\eta^{\left(l\right)}\left[n+1\right]-d_{x}^{\left(l\right)}\left[n\right]-\eta^{\left(l\right)}\left[n\right]\right)}^{2} \end{array}$$
(24e)$$\begin{array}{cc} \; & +{\left(d_{y}^{\left(l\right)}\left[n+1\right]+\xi^{\left(l\right)}\left[n+1\right]-d_{y}^{\left(l\right)}\left[n\right]-\xi^{\left(l\right)}\left[n\right]\right)}^{2}\leq v^{2},\;n=1,\ldots ,N-1; \end{array}$$
(24f)$$\begin{array}{cc} \; & {\left(d_{x}^{\left(l\right)}\left[N\right]+\eta^{\left(l\right)}\left[N\right]-L\right)}^{2}+{\left(d_{y}^{\left(l\right)}\left[N\right]+\xi^{\left(l\right)}\left[N\right]\right)}^{2}\leq v^{2}. \end{array}$$
By combining Proposition 1 and Proposition 2, the information causality constraint ([Disp-formula FD9a-sensors-19-03442]) in Problem P1 becomes ([Disp-formula FD24a-sensors-19-03442]). Constraints ([Disp-formula FD24b-sensors-19-03442])–([Disp-formula FD24c-sensors-19-03442]) and the additional variables ${\left\{R_{d}^{\left(l+1\right)}\left[n\right],R_{e}^{\left(l+1\right)}\left[n\right]\right\}}_{n=1}^{N}$ are added to the optimization problem to guarantee validity and convergence of the proposed lower bounding solution approach.

It can be shown that the support of the variables is a convex set and the second-order derivatives of all function and constraints are positive semidefinite. As a result, Problem $P2^{\left(l\right)}$ for the *l*th iteration is a convex problem, which can be readily solved by standard convex optimization solvers such as CVX [30].

The proposed iterative UAV trajectory optimization algorithm for secure UAV mobile relaying is summarized in Algorithm 1.

**Algorithm 1** The iterative UAV trajectory optimization algorithm
1:Initialize the UAV relay’s trajectory ${\left\{\left(d_{x}\left[n\right],d_{y}\left[n\right]\right)\right\}}_{n=1}^{N+1}$, with fixed starting and ending points $(d_{x}\left[1\right],d_{y}\left[1\right])=\left(0,0\right)$ and $(d_{x}\left[N+1\right],d_{y}\left[N+1\right])=\left(L,0\right)$. Let the initial iteration count l=0.2:
**repeat**
3:Find the optimal solution ${\left\{\eta^{\left(l\right)}\left[n\right],\xi^{\left(l\right)}\left[n\right]\right\}}_{n=1}^{N}$ to Problem $P2^{\left(l\right)}$.4:Update the trajectory anchor points as $d_{x}^{\left(l+1\right)}\left[n\right]=d_{x}^{\left(l\right)}\left[n\right]+\eta^{\left(l\right)}\left[n\right]$ and $d_{y}^{\left(l+1\right)}\left[n\right]=d_{y}^{\left(l\right)}\left[n\right]+\xi^{\left(l\right)}\left[n\right]$.5:Set l=l+1.6:**Until** terminate at convergence or a predefined maximum number of iterations is reached.


## 4. Numerical Results

In this section, simulation results are presented to verify the proposed iterative trajectory optimization technique for secure buffer-aided UAV mobile relaying. The UAV-assisted mobile relaying system model as shown in Figure 1 is adopted. The starting point **SP** and the end point **EP** of the trajectory are designated as the origin and (0,L), respectively. The source and the destination are located at fixed points $\left(L_{s},0\right)$ and $\left(L_{d},0\right)$ on the horizontal axis. The eavesdropper location follows a uniform distribution between [a,b] on the horizontal axis. The UAV relay moves in the upper half-plane of the 2D space, i.e., $d_{y}>0$, at height *h* above the terrestrial communication system. Simulation parameters are summarized in the following Table 1.

Among them, v,a,b are adjusted in the simulations to observe their impacts on the system performance.

### 4.1. Convergence of the Secrecy Rate Performance

First of all, we investigated how the average ergodic secrecy rate of the buffer-aided UAV mobile relaying system achieved by the proposed iterative optimization scheme changes with the number of trajectory iterations and the UAV relay’s maximum speed. The boundaries of the uniformly distributed eavesdropper location were a=300 m and b=500 m. The total flight time was set to 80 s. Several maximum UAV speed values v=16 m/s, v=18 m/s, and v=20 m/s were examined. The simulation results (average secrecy rate versus iteration number) are shown in Figure 2. The average secrecy rate curve of a system without trajectory optimization is shown as the solid line without marks in the figure to provide a performance benchmark.

It can be observed from Figure 2 that as the proposed trajectory optimization algorithm iterates, the overall average secrecy rate increases. The performance achieved by the proposed algorithm converged very fast in the first two to three iterations, and became levelled off in less than 10 iterations for all the scenarios examined. The performance achieved by three algorithm iterations was over 99% of that at convergence (10 iterations). Because the subproblem in each iteration is strictly convex, which can be readily solved by a classic convex optimization algorithm, the complexity of each algorithm iteration is almost fixed. The overall complexity of the proposed iterative optimization algorithm is mainly determined by how fast the iterative procedure converges. The proposed iterative algorithm is therefore practically desirable because the numerical study revealed that near optimal solutions can always be obtained in a small number of (around 3) iterations. The fast convergence property then indicates relatively low complexity of the proposed algorithm in practical implementations. This is desirable from both theoretic study and practical system design perspectives. It is also observed that higher maximum UAV speed is beneficial to the system’s overall secrecy rate. Increasing *v* from 16 m/s to 20 m/s resulted in over 9% improvement to the average secrecy rate. This is because the greater the maximum UAV speed, the less constrained the trajectory. A more favorable trajectory that achieves greater secrecy rate can be obtained accordingly. Obviously, the greater the number of iterations, the closer the trajectory to the optimal. That means as the trajectory is updated in the proposed iterative procedure, it is gradually optimized and eventually converges to the optimal trajectory.

How the location of the eavesdropper impacts the overall average secrecy rate performance was studied by examining different boundary values *a* and *b* for the uniform distribution. Maximum UAV speed v=20 m/s and total flight time T=80 s were used in this part of simulation. Simulation results for 5 different [a,b] combinations are presented in Figure 3.

For all the scenarios examined, the overall average secrecy rate increases as the trajectory optimization algorithm iterates, and fast convergence as in Figure 2 can also be observed. The eavesdropper location further away from the destination (closer to the source) is shown to be beneficial to the overall average secrecy rate performance. This is mainly because when the first hop communication is completely obstructed on the ground, the forwarded signal from the UAV relay is the only source of information leakage to the eavesdropper. It is, therefore, not desirable to have an eavesdropper closer to the destination such that the R-D and R-E channels are more correlated, which violates the basic principle for PHY security design.

### 4.2. Trajectory Regarding Iteration Number and Eavesdropper Location Distribution

We next present the obtained UAV trajectory in the 2D space to show how the optimized trajectory is approached as the number of algorithm iterations increases. The impact of eavesdropper location on the optimized UAV trajectory was also investigated. In this part, the total flight time was set to T=80 s, and the maximum UAV speed was v=16 m/s. An eavesdropper uniformly distributed between [300,500] on the $d_{x}$ axis was considered to demonstrate the iterative update process of the UAV trajectory. It is observed in Figure 4 that as the proposed algorithm iterates, the UAV’s trajectory gradually converges. Convergence of the trajectory was achieved at about 10 iterations, which validates the effectiveness of the proposed algorithm in following an optimized trajectory.

In Figure 5, the optimized UAV trajectories obtained by 12 algorithm iterations are presented for three groups of eavesdropper locations. The selection of 12 iterations was based on the authors’ observations from the numerical studies (as shown in Figure 2 and Figure 3), which guaranteed to give the converged trajectory. It can be observed that when the eavesdropper location is further away from the destination (closer to the source), the UAV’s optimized trajectory has a shorter total flight distance. That means with fixed flight time, if the eavesdropper is expected to be closer to the destination, the UAV needs to fly faster to create a trajectory that can avoid potential eavesdropping as much as possible. As a result, having an eavesdropper far away from the destination is beneficial to both the sum secrecy rate performance and the energy efficiency.

## 5. Conclusions

The trajectory optimization problem for PHY security of a buffer-aided UAV mobile relaying system with a randomly located eavesdropper has been studied. The problem of optimizing the anchor points of the discretized piecewise linear trajectory for maximized sum secrecy rate under information causality and maximum UAV speed constraints has been formulated and shown to be non-convex. By changing the optimization variables to the iterative trajectory increments on each anchor point and invoking a lower bounding technique for the achievable rates, the problem has been reformulated and decomposed into a series of convex optimization subproblems through an iterative procedure. Based on the squeeze principle, convergence of the iterative optimization approach has been achieved by adding extra upper bound constraints to the achievable rates. This successive convex approximation procedure is shown to approach the optimal trajectory progressively with good convergence property. The optimality gap between the approximate convex problem and the original non-convex problem has been shown to be very small with only a few (about 3) iterations. The complexity of the proposed iterative optimization algorithm is thus practically low. The optimal PHY secure UAV relay trajectory has been obtained through the iterative procedure after a few iterations. It has been observed from the simulation results that higher maximum UAV speed would improve the sum secrecy rate performance because it gives higher flexibility to the trajectory. The simulation results have also revealed that an eavesdropper further away from the destination is beneficial to both the sum secrecy rate performance and the UAV relay’s energy efficiency.

## Figures and Tables

**Figure 1 sensors-19-03442-f001:**
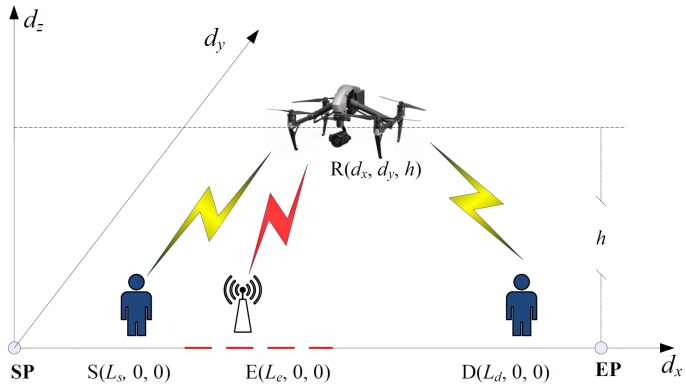
The UAV-enabled mobile relaying system model.

**Figure 2 sensors-19-03442-f002:**
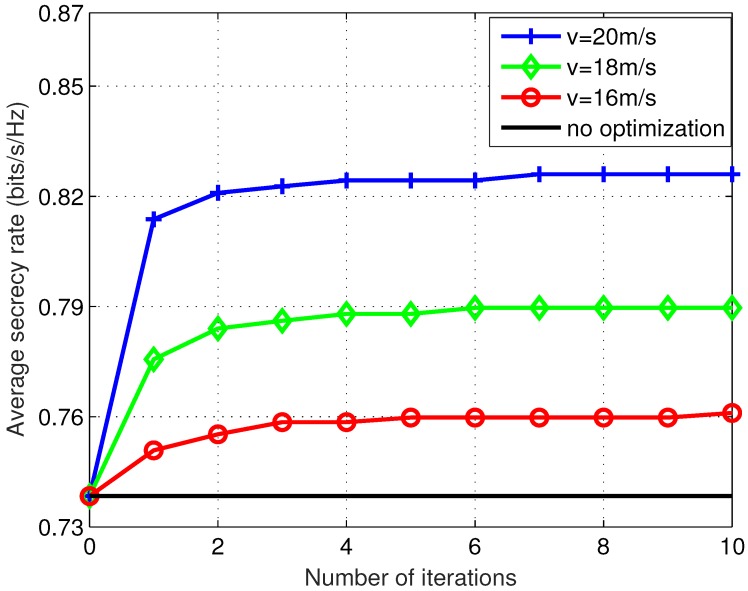
Convergence of the average secrecy rate performance with different maximum UAV speed values. The total flight time is T=80 s. The proposed algorithm exhibits fast convergence property in all the scenarios examined. Higher maximum UAV speed results in higher average secrecy rate performance.

**Figure 3 sensors-19-03442-f003:**
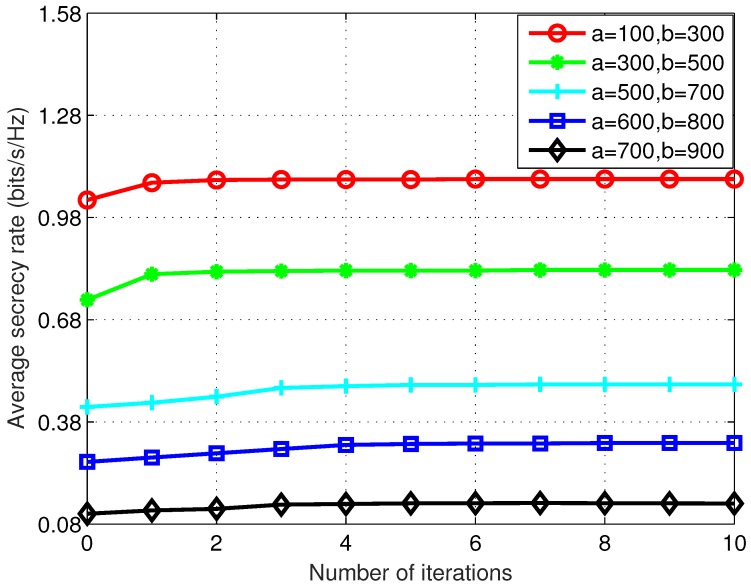
Average secrecy rate performance for different distribution boundaries of the eavesdropper location with maximum UAV speed v=20 m/s and total flight time T=80 s. The eavesdropper located further away from the destination is shown to be more favorable to the overall average secrecy rate performance.

**Figure 4 sensors-19-03442-f004:**
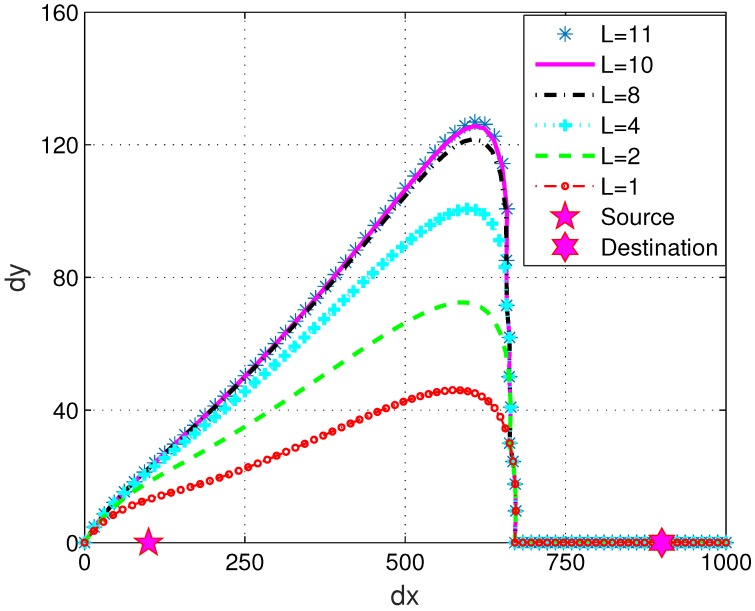
The iterative updates of the UAV trajectory with maximum UAV speed v=16 m/s and total flight time T=80 s. The eavesdropper location is uniformly distributed between [300,500] on the $d_{x}$ axis. The UAV’s trajectory converges in about 10 iterations.

**Figure 5 sensors-19-03442-f005:**
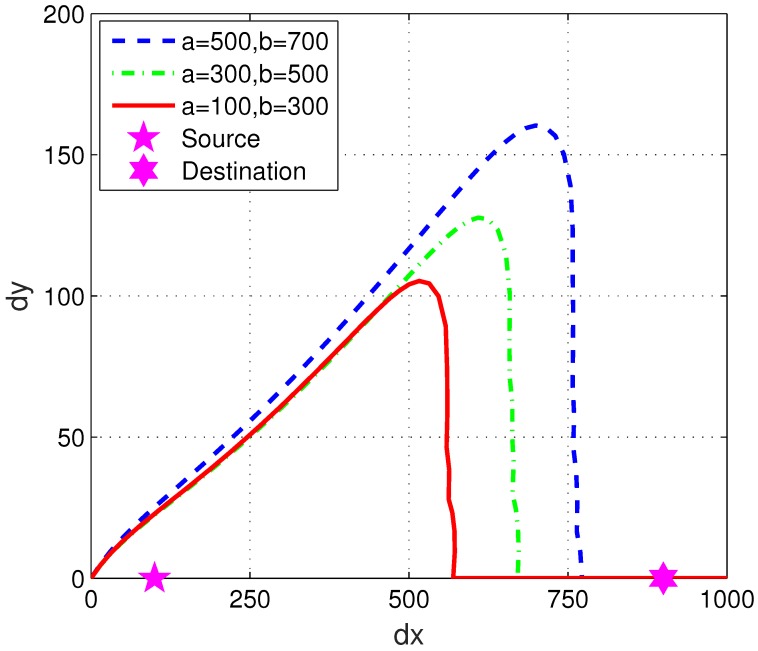
The optimized trajectories for different eavesdropper locations with maximum UAV speed v=16 m/s and total flight T=80 s. It is observed that when the eavesdropper location is further away from the destination, the UAV’s optimized trajectory has a shorter total flight distance, which is both spectrum-efficient and energy-efficient.

**Table 1 sensors-19-03442-t001:** Values of the System Setting Parameters for Simulation.

Parameter	Value
Height of UAV trajectory *h*	100 m
**SP**-to-**EP** distance *L*	100 m
Location of the source $L_{s}$	100 m
Location of the destination $L_{d}$	900 m
Transmit power of the source and the relay $p_{s}$, $p_{r}$	20 dBm
Power spectral density of AWGN	−174 dBm/Hz
Bandwidth	10 MHz
Total flight time *T*	80 s
Adjustable parameters	v,a,b

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
