# Peer review of "Iterative Trajectory Optimization for Physical-Layer Secure Buffer-Aided UAV Mobile Relaying"

_sensors, 2019, doi:10.3390/s19153442_

Round 1

Reviewer 1 Report

The reviewer has no comments and the paper can be accepted as it is. 

Reviewer 2 Report

I congratulate the authors for doing a thoughtful job with the revision. My comments have been carefully taken into account, and I believe the paper is ready for publication.

This manuscript is a resubmission of an earlier submission. The following is a list of the peer review reports and author responses from that submission.

Round 1

Reviewer 1 Report

The authors study how to design trajectories of UAVs, so that their role as relays can be maximally efficient in scenarios with potential privacy concerns. The problem is really interesting and timely, as UAV technology and data privacy are both hot topics nowadays. Although the question addressed is very interesting, my main concert is about a number of modeling assumptions. In the following, I will first present my main concerns, and then list some minor observations.

Main comments:

1. There are an important number of assumptions that are not well motivated. The authors just mention that they are taking them, but it would be desirable if they could also explain the reasoning behind these choices. In particular:

a) why sender, received and eavesdropper are all co-linear? in principle the eavesdropper could be anyway, and not only in the same line defined by sender and receiver.

b) the eavesdropper location is assumed to be uniformly distributed in a segment [a,b]. How to choose a,b? do they affect the results? why to assume that the location is Gaussian, or some other unbounded distribution?

c) why is the relay using TDMA? could it use FDMA, CDMA, or maybe a full-duplex technology?

2. The channel model -- described in Eq. (2) -- is only valid for high-altitudes, where there is LoS condition and hence path-loss exponent = 2. However, if the UAVs are at low altitudes the path-loss exponent get affected continuously, until at very low heights one recovers the ground-level path-loss (please see e.g. [1]).

[1] Azari, Mohammad Mahdi, Fernando Rosas, Kwang-Cheng Chen, and Sofie Pollin. "Joint sum-rate and power gain analysis of an aerial base station." In 2016 IEEE Globecom Workshops (GC Wkshps), pp. 1-6. IEEE, 2016.

3. I find Eq. (8) a quite delicate choice, which is not discussed with the care it needs. Why is it appropriate to consider the mean value? For example, many security design is done considering the worst-case-scenario, i.e. considering the max instead of the mean.

4. I wonder about the condition of causality in the optimization problem. To make it truly causal, wouldn't it be necessary to have condition (24a) to hold not only for the sum until N, but for the sum until all j=1,..., N?

5. After presenting a very interesting problem, I find Sections 4 and 5 a bit weak. This is a theoretical paper, and hence the most interesting result (in my humble opinion) are insights, which could be latter exported to enable new designs in real-world applications. At this respect, a number of interesting findings are reported, but they are not sufficiently explained. For example, it is said that having a higher max speed allows for better optimization, but it is not described the kind of trajectory that it enables. Other example: it is said that the iterations needed for convergence are usually few, but why is this interesting? it is not clear if the UAV themselves are supposed to compute the optimal path, and if that is the case if their computational capabilities is such that such computation with many interations could be somehow challenging. Couldn't this computation be done in a central computer anyway? Please develop the findings further, and try to provide insights that could be of interest for future UAV designs.

6. The captions in most figures are mere descriptions of the graphs. Please use them to convey a message, which may provide insightful  descriptions of the results. 

Minor observations:

7. It would be important for the readers to get a sense of how useful -- or limited -- is the approach of information-theoretic privacy in real life applications.

8. In Figure 1, why not to use "d_z" to denote the third axis? "h" is already used for the variable that denote the altitude of the UAV.

9. When introducing the secrecy rate, please provide more background to the uninformed reader. If my memory is correct, the secrecy rate in the general case is described by a quite complicated expression, which finds a simpler expression for degraded eavesdropper cases and also Gaussians channels... is that correct? It could be useful for the average reader to get a quick recap on where this expression is coming from.

10. When writing the speed constraints, it could be done in a single equation by taking the appropriate values for d[0] and d[N]. Please do so, to simplify the statement of the optimization problem.

11. I think the presentation of Algorithm 1 could be improved. Try to make it a bit more formal.

12. In Table 1, the symbol (e.g. "h" for height) should be listed in the first column, after the full name and in parenthesis.

13. Do the height of the UAV affect the results? Could it be possible / interesting to find the height that maximizes the secrecy? 

Reviewer 2 Report

This paper studied the trajectory design in UAV enabled physical layer scenario. Overall the paper is well written and quite easy to follow. The reviewer has the following comments.

1. The considered scenario is a sub-system in the following work, which has provided a comprehensive reviewer of UAV communications from the physical layer perspective. It is better to discuss it.

“Safeguarding Wireless Networks with UAV: A Physical Layer Security Perspective,’’ IEEE Wireless Communications, 2019. https://arxiv.org/pdf/1902.02472.pdf

2. In the paper, only one eavesdropper is considered, which may not be practical. Is it possible to extend the result to the multi-eavesdropper case? In addition, the eavesdropper’s location is assumed to be known, how can this be possible in practice? The following work has considered a more general case without knowing the exact locations of eavesdropper.

“Robust Trajectory and Transmit Power Design for Secure UAV Communications,’’ IEEE Transactions on Vehicular Technology, vol. 67, no. 9, pp. 9042-9046, Sep. 2018.

3. Why the LOS channel is assumed? In practice, it is more likely to have the Rician channel due to the high-rise building and trees.

4. It is better to discuss the complexity of the proposed algorithm.

5. Some practical issues are ignored, such as the user delay and the multiUAV scenario. For example, how to jointly design the trajectory if multiUAVs are used? Or the user at the destination has the delay sensitive service, which is one of the main cases in 5G network. The following works may be helpful. 

 “Common Throughput Maximization in UAV-Enabled OFDMA Systems with Delay Consideration,’’ IEEE Transactions on Communications, vol. 66, no. 12, pp. 6614-6627, Dec. 2018.
